# Phylogeography of Sub-Saharan Mitochondrial Lineages Outside Africa Highlights the Roles of the Holocene Climate Changes and the Atlantic Slave Trade

**DOI:** 10.3390/ijms23169219

**Published:** 2022-08-16

**Authors:** Luísa Sá, Mafalda Almeida, Simon Azonbakin, Erica Matos, Ricardo Franco-Duarte, Alberto Gómez-Carballa, Antonio Salas, Anatóle Laleye, Alexandra Rosa, António Brehm, Martin B. Richards, Pedro Soares, Teresa Rito

**Affiliations:** 1CBMA (Centre of Molecular and Environmental Biology), Department of Biology, University of Minho, 4710-057 Braga, Portugal; 2Institute of Science and Innovation for Bio-Sustainability (IB-S), University of Minho, 4710-057 Braga, Portugal; 3Laboratory of Histology, Reproductive Biology, Cytogenetic and Medical Genetics, Faculty of Health Sciences, University of Abomey-Calavi, Cotonou 00229, Benin; 4Grupo de Investigacion en Genetica, Vacunas, Infecciones y Pediatria (GENVIP), Hospital Clínico Universitario, Universidade de Santiago de Compostela, 15706 Santiago de Compostela, Spain; 5GenPoB Research Group, Instituto de Investigación Sanitaria (IDIS), Hospital Clínico Universitario de Santiago (SERGAS), 15706 Santiago de Compostela, Spain; 6Unidade de Xenética, Instituto de Ciencias Forenses, Facultade de Medicina, Universidade de Santiago de Compostela, 15706 Santiago de Compostela, Spain; 7Human Genetics Laboratory, Campus da Penteada, University of Madeira, 9020-105 Funchal, Portugal; 8Faculty of Life Sciences, Campus da Penteada, University of Madeira, 9020-105 Funchal, Portugal; 9Department of Biological and Geographical Sciences, School of Applied Sciences, University of Huddersfield, Huddersfield HD1 3DH, UK

**Keywords:** phylogeography, mitochondrial DNA, founder analysis, Holocene, slave trade influence, computational approach

## Abstract

Despite the importance of ancient DNA for understanding human prehistoric dispersals, poor survival means that data remain sparse for many areas in the tropics, including in Africa. In such instances, analysis of contemporary genomes remains invaluable. One promising approach is founder analysis, which identifies and dates migration events in non-recombining systems. However, it has yet to be fully exploited as its application remains controversial. Here, we test the approach by evaluating the age of sub-Saharan mitogenome lineages sampled outside Africa. The analysis confirms that such lineages in the Americas date to recent centuries—the time of the Atlantic slave trade—thereby validating the approach. By contrast, in North Africa, Southwestern Asia and Europe, roughly half of the dispersal signal dates to the early Holocene, during the “greening” of the Sahara. We elaborate these results by showing that the main source regions for the two main dispersal episodes are distinct. For the recent dispersal, the major source was West Africa, but with two exceptions: South America, where the fraction from Southern Africa was greater, and Southwest Asia, where Eastern Africa was the primary source. These observations show the potential of founder analysis as both a supplement and complement to ancient DNA studies.

## 1. Introduction

Whilst the longer-term emergence of Homo sapiens was likely the result of “multiregional” processes across sub-Saharan Africa [1,2,3,4,5,6], fossil cranial morphology, contemporary genomes and ancient genome-wide variation sampled over the past 18,000 years all point to a tripartite population structure within sub-Saharan Africa [7,8,9,10]. Such population structure had previously been recognised by maternally inherited mitochondrial DNA (mtDNA) genome analysis that established a chronology for the settlement of such groups through migrations that occurred 100,000–200,000 years ago (100–200 ka) [4,5,6], determining three groups of anatomically modern humans (AMH) in Central/West, Eastern, and Southern Africa that are clear at the phenotypic and genome-wide level [7,8,10]. A recent study using ancient genomic data suggests that by ~5 ka a cline existed between these three sources of modern human ancestry [10].

On the mtDNA phylogenetic tree, the first major split in the extant variation, dating to nearly 200 ka using various approaches [4,5,6,11], occurred between macrohaplogroup L0, typically found in indigenous Southern African “Khoe-San” click-speakers, and the remaining L1′2′3′4′5′6 (or L1′6 for short [12]) branches distributed across populations further north. This primary south–north divergence has also been detected at the genomic level [7,8,10,13].

On the northern side, African haplogroup L1 split first from L2′3′4′5′6 ~150 ka, representing Central and Eastern African clades, respectively, followed by L5 within Eastern Africa ~130 ka. While L2′3′4′6 diversified mainly in Eastern Africa, L2 split from L3′4′6 into West Africa after ~100 ka via a route that is not completely clear [14]. About 60–75 ka, L3 arose in Eastern Africa and expanded dramatically prompted by the population growth of Eastern African populations. This may have been fuelled by the moister conditions following a major drought [15,16] and the possible arrival of novel uses of symbolic activities and technology associated with aspects of “modern” behaviour that took hold in southern African refugia ~130–75 ka [17]. This is suggested by major dispersals of L0 lineages from Southern to Eastern Africa occurring at this time [6] and may have led to the establishment of long-term networks between South and East Africa [18]. As a result of these expansions centred on Eastern Africa, L3 is today well represented across Africa, including in deep clades in Central and West Africa, and also encompasses all non-African mtDNA variation, following dispersals out of Eastern Africa into Eurasia that took place as part of the same process [16,19]. Reciprocal exchanges are also evident, for instance, haplogroup L2 lineages also show signs of being transmitted back from West/Central to Eastern African people around this period [14]. 

The following 40,000–50,000 years (40–50 ky) witnessed little significant population movement within Africa and variation accumulated regionally, likely due to a fragmented environment [15]. However, the climatic changes introduced during the Holocene (with the peak of the Holocene Climate Optimum ~8 ka) allowed woodland and savanna adjacent to the Sahara to expand [20], creating conditions for the Sahel to become a major corridor, both north–south and east–west [21,22,23]. The expansion of L3 lineages in Eastern Africa and L2 lineages in West Africa attest to the rise in human mobility across the Sahel during this period of improved climate [4]. Movements northwards of eastern, central, and western populations also led to the arrival of sub-Saharan mtDNA lineages in North Africa and, eventually, Europe [24], which was most likely the largest movement of people from Sub-Saharan Africa into Europe and Asia since the out-of-Africa migration. By the end of this period, Africa might have also witnessed the expansion of pastoralism across the northern range of the continent [25,26].

From ~4 ka, another major expansion substantially shifted the genetic structure of the African continent—the dispersal of Bantu speakers. Driven by the mastering of agriculture and iron-smelting techniques, these migrations began between Nigeria and Cameroon and followed two main routes into Southern Africa: (i) a western route directly into Angola, Botswana, and South Africa around 3.5 ka [27,28], and (ii) an eastern route into the Great Lakes in Uganda at 2.5 ka [29] and from there southward into Mozambique, less than 2 ka [5,14,30,31,32]. All these demographic events over the last millennia have reshaped the earlier genetic diversity and established the current population genetic structure within Africa, which is phylogeographically well studied (reviewed in [33]).

In the last 500 years, major forced migrations from Africa into other parts of the world have occurred. While slavery had been present in human societies for thousands of years, the scale of the Atlantic slave trade was unparalleled. From the 15th to the 19th centuries, more than 10 million Africans were carried into Europe and, especially, into the Americas [34]. The European slave trade started in West Africa in the 15th century, in the region then known as Senegambia, between the Senegal and Gambia rivers [35]. In the ensuing years, following the settlement of European colonies along the west coast of Africa, other regions became more important sources of slaves including Mina, the current Ghana/Benin region. Historical records suggest that these regions contributed up to two-thirds of slaves, followed in the 17th century by Angola, located further south [36]. The precise origin and ethnicity of the people forced to travel is not well documented, although a general overview of dominant ethnicities exists [37], and a genetic picture has begun to emerge from both mtDNA [38,39,40] and on a genome-wide level [32,41,42]. 

Phylogeographic analysis is useful to shed light on unknown aspects of migrations that occurred across the African continent. This is a methodology that employs the phylogenetic reconstruction of non-recombining genomic loci and the nesting of lineages coupled with geographic information to infer the timing and the direction of migrations [43]. Founder analysis, in turn, is a formalization of the phylogeographic methodology applicable when a source and sink population can be established [43,44,45,46]. The clear African origin of human mtDNA lineages offers an opportunity to test the accuracy of the founder analysis approach using a controlled case scenario, such as the long-distance gene flow mediated by the Atlantic slave trade into North America, and to simultaneously investigate both the timing and the magnitude of other, less well understood dispersal events into Europe and Asia and the source of their founder lineages within Africa.

## 2. Results

Following the data collection, alignment and quality control during phylogenetic analysis, we obtained a total of 6048 mitogenomes, whose identification, location, and haplogroup classification are displayed in Appendix A. We performed founder analysis using six scenarios, aiming at either the validation of the methodology or at uncovering specific aspects of past population movements involving African mitogenome lineages. A list of founders is displayed in Appendix A, totalling 74 founders for North Africa, 114 founders for Southwest Asia, 78 founders for Europe, 51 founders for Iberia, 285 founders for North America and 25 founders for South America. The results of the scans for the six models of founder analyses across the last 25 ky are displayed in Figure 1.

For the founder analyses of sub-Saharan lineages into North Africa (A), and for both North and sub-Saharan lineages as the source into Southwest Asia (B), Europe (C), and Iberia alone (D), we obtained striking results. The migration scans indicate two main peaks of migration, one in the early Holocene around the Holocene Climate Optimum (~7.4 ka–10 ka), and another in the present or close to present (~0 ka). We see an absence of movements carrying sub-Saharan lineages between those periods. As expected, for North and South America, we see only recent episodes of gene flow. 

Overall, the clarity of the patterns highlights the strength of the analyses performed. However, some small bumps at about 1.0 ka most likely represent arrivals with the Atlantic slave trade, possibly conflated with a few earlier minor movements in the case of Europe, and probably artificially inflated due to the presence of a small number of sequencing artefacts, which would have a disproportionate impact at this timescale. It is also possible that the mutation rate correction for purifying selection [47] at this very recent time depth is insufficient. Given the high number of founders detected in North America, it is even possible that this number is still underestimated since while the main lineages were detected, some of the variation within those lineages might have already been carried to North America, therefore slightly increasing the estimated founder ages. Further sampling of African populations (especially West African populations) should help to settle this issue.

Making this assumption, and given the above results, we performed a second analysis, stipulating a model with two migration events. The first is at 8 ka, corresponding to the Holocene Climate Maximum, accounting for the 7.4–10.0 ka peaks in the founder analyses into North Africa (A), Southwest Asia (B), Europe (C) and Iberia (D), and the second is at 0.5 ka, corresponding roughly to the start of the Atlantic slave trade, which will statistically allocate any events within the last few millennia. The founder lineages identified can then be further analysed phylogeographically for a better definition. 

Figure 2 displays the results of the founder analysis using this two-migration model, respectively labelled as “prehistoric” and “historical” (Figure 2A). It is important to note that, in general, founder lineages are statistically allocated to one migration or the other, with little evidence of gene flow in between, as suggested by the scan. Furthermore, we verified the phylogenetic tree structure and phylogeography for each founder to pinpoint their most likely African broad region of origin (Figure 2B).

The results for North Africa suggested that about half of the sub-Saharan lineages arrived in the early Holocene, and the other half in recent times. However, when analysing the putative source location of the lineages in each migration event, the distribution differed. Amongst the prehistoric arrivals, half of the North African mtDNA founders were from West Africa, with much of the other half associated with Eastern Africa. However, there were substantially more lineages from West Africa, and correspondingly fewer from Eastern Africa, when considering the origin of lineages in recent times.

Again, the present-day sub-Saharan lineages in Southwest Asia are distributed nearly 50:50 between prehistoric and historical arrivals, although with different distributions in present-day populations (Figure 3). The phylogeographic analysis showed that the majority (86%) of the African lineages arriving in Southwest Asia in the prehistoric period were from Eastern Africa. The exchange of female lineages between Eastern Africa and Southwest Asia (mostly Arabia) in the postglacial period, from 15–7 ka, has been documented before [4,48] and is supported again here, displaying up to a frequency of 4–10% in current populations (Figure 3A). Curiously, for the recent arrivals, which reach frequencies of up to 13% in Arabia (Figure 3B), although the majority are again from Eastern Africa (68%), there is a significant fraction of lineages arriving from West Africa (22%). 

For Europe, our founder analyses suggest that there are more African mtDNA lineages related to the prehistoric than the historical period (~60:40). Note, however, that most of the lineages here considered in the European analysis are from Iberia, as displayed in Figure 3 for both prehistoric (Figure 3A) and historical arrivals (Figure 3B). Typically, North African lineages are well represented in the prehistoric period (a third of the total). However, the higher percentage of sub-Saharan lineages arriving in Europe during this period, mostly from West Africa (63%) is likely from migrants that first crossed the Sahel belt into North Africa and moved from there into Europe via the Strait of Gibraltar. Although not detected in modern North Africa’s maternal genetic background (which is very heterogeneous: [49]), these results suggest that most early Holocene migrations from Africa into Europe occurred via the West Mediterranean. 

When considering historical migrations, although gene flow across the Mediterranean must again have played an important role, the North African influence diminishes, with West African lineages becoming more dominant representing three-quarters of the migrants. This suggests that a higher percentage of recent migrations from West Africa into Iberia and Europe took place without crossing North Africa. Furthermore, some lineages arrived directly from Southern Africa (~9% in Europe and ~9.5% in Iberia), without being detected either in North or in West, Central or Eastern Africa. We did not detect this for arrivals in the early Holocene and this likely reflects the more extensive sea crossings of the last five centuries or so.

For the founder analysis of maternal lineages into North and South America, we maintained the two migration events; this allowed testing of the discriminatory capacity of the methodology. Our analysis allocated 6% of North American and 3% of South American sub-Saharan lineages to the Holocene. This could be explained by statistical residuals from the recent lineages, but also from a couple of lineages whose founders in Africa were likely not detected, or due to minor errors in the sequences leading to overestimates of the age estimate of specific lineages. Nevertheless, even considering these sources of error, it is clear that only recent migrations mainly associated with the transatlantic slave trade are significant. 

As for recent mtDNA lineages in North America, mostly labelled as African-Americans in the databases, more than three-quarters are originally from West Africa, with nearly 15% arriving from Southern Africa. We see a striking contrast between North and South America. South America shows more than half of the lineages (~55%) arriving from Southern Africa (mostly Angola, from the phylogeographic inferences), against ~45% from West Africa, pinpointing an increased importance of the Southern African people being transported to South America.

## 3. Discussion

In this work we estimated the timing, the proportion, and the broad geographic origin of relevant movements of African mtDNA lineages outside of sub-Saharan Africa, using a phylogeographic-based approach and, more specifically, the founder analysis methodology [45]. Previous inferences regarding the presence of sub-Saharan mitogenomes in Europe have highlighted movements from North Africa, following the Ice Age and with the onset of the Holocene in Africa ~8–9 kya [24], including gene flow specifically to Iberia [50]. We were able to contrast the proposed Holocene event with a recent and well-known event, namely the slave trade and eventually more recent migrations, and quantify the impact of each on the maternal gene pool. The founder analysis scans into North Africa, Europe and Southwest Asia all showed two major periods of migration: one in the early Holocene and a much more recent one. Thus, these movements were not random but rather concentrated in specific time periods. 

Following the “out-of-Africa” migration ~60–70 kya, significant numbers of sub-Saharan lineages did not disperse from the continent again until the early Holocene. At this time, the climate optimum and the consequent expansion of equatorial forest allowed movements that had previously been limited. Populations were able to cross a greener Sahara to North Africa and Southwest Asia, and from there into Europe, as suggested before [24,50], demonstrating the continued role of climatic oscillations in shaping continental genetic structure. The genetic input from sub-Saharan populations into North Africa points to two major sources. One is West Africa, reinforcing the trans-Saharan movements made possible by the more favourable environmental conditions. The second is Eastern Africa, a likely source of pastoralism in this period [21,25]. This is an example of the improved climate also allowing populations with different lifestyles to prosper, with pastoralists spreading from Eastern to both West Africa and North Africa [21,25,26]. 

When considering more recent periods, migrations from West African lineages dominate most of the mtDNA founder analyses. For Europe and Iberia specifically, North Africa decreases in relevance as a source, suggesting that a higher percentage of lineages arrived there without crossing the Sahel into North Africa. During this period, lineages arriving from Southern Africa (mostly Angola), another area with prominent slave-trade ports, are also detected, a scenario not observed for the early Holocene. Therefore, although some dates might be over-estimated, most of these lineages are likely to have their source in the Atlantic slave trade moving into Europe by maritime transport.

Even in North Africa and in Southwest Asia, far from the direct influence of the Atlantic slave trade, there is a rise in the arrival of West African lineages in recent times, corresponding to nearly a quarter of their sub-Saharan lineages. This might be due to other networks of human traffic that also forcibly took people for labour, which were in operation at least several millennia earlier. For example, the trans-Saharan slave trade went back at least into the Roman period [51] and has previously been identified as being responsible for mitogenomes seen in modern Near Eastern and Arabian populations [52].

The Atlantic slave trade left clear signals of African maternal gene flow in both Europe and the Americas [38,39]. It started about 500 years ago and, during its earlier stages, the bulk of people captured for slavery were from West Africa. The West African signal was, therefore, by far the most important African genetic imprint in Europe and North America. The southern African signal in North America corresponds mainly to Angola, with no discernible signal from South-Eastern Africa, which is in line with earlier studies using less discriminate mtDNA control-region data [39]. Nevertheless, we also detected a small number of founders tracing to Eastern Africa that could nevertheless represent a distinct and more recent migration from this region or adjacent parts of Southwest Asia. However, in our founder analysis, while West Africa dominated the signal for North America, we could discern a different pattern in South America, where southern African mitogenomes (mostly from Angola) are found at a slightly greater percentage than West African founders. 

With the Atlantic slave trade, by the 17th century, ports were established to the south of West Africa, in Angola [36]. Our results indicate that whilst this region contributed only a small fraction to the number of slaves forcibly taken into North America, its impact on South America was much greater. This corroborates the results from genome-wide analysis by Gouveia et al. [53] and the identity-by-descent approach of Micheletti et al. [54], which also suggested a substantially higher percentage of Southern African ancestry in South Americans of African descent.

## 4. Materials and Methods

We collected a database of all published mitogenomes classified as sub-Saharan African L0 to L6 haplogroups. We also added the recently sequenced mitogenomes from Iberia published meanwhile [55]. We aligned all sequences against the revised Cambridge Reference Sequence (rCRS) with Mafft (Version 7, https://mafft.cbrc.jp/alignment/server/ accessed on 28 January 2021) [56] and manually checked problematic stretches of mtDNA. We classified samples into haplogroups using Haplogrep 2 [57] and grouped them into major subclades L0, L1, L2, L3, L4, L5, and L6 for phylogenetic analysis. We performed phylogenetic reconstruction using the Network software, using the reduced-median algorithm [58] and weighting mtDNA sequence positions according to their relative frequency [47], guided by the tree structure provided by PhyloTree [59]. We calculated age estimates of all tree nodes using rho and the time-dependent molecular clock established by Soares et al. [47].

Following the phylogenetic reconstruction, we applied founder analysis with software we have been developing [60]. The software (which is under development) reads the entire phylogenetic tree in an .xml format. It searches for shared haplotypes between the hypothetical source and the hypothetical sink (either existing or inferred ancestral haplotypes). Following the detection of hypothetical founders, certain conditions need to be fulfilled to be considered a founder, as stipulated in its original conception [46]. We here applied an *f*1 criterion, meaning that a founder match must display some diversity in the source population, as used before [5,16,44,45,46,61]. We considered six models in the founder analysis. In the first case, sub-Saharan Africa was considered the source for North Africa (A), and in the remaining cases, we considered all of Africa as the source into Southwest Asia (B), Europe (C), Iberia alone (D), North America (E) and South America (F). To perform an unbiased evaluation of probable migration times, we estimated migrations every 200 years from 0–25 ka. Using this approach, we performed a scan on the statistical distribution of the lineages across a linear timeframe allowing the detection of the most probable periods of migration given the data. Such an approach should be further validated and improved using other sources of information (namely from archaeology, history, paleoclimatology and linguistics), but the present study marks a step forward in this regard.

We inferred migration times for each founder using rho, which has proven to be an unbiased estimator [62]. We calculated standard errors according to Saillard et al. [63], allowing the estimation of realistic errors in the Bayesian partitioning, as introduced before in the methodology [16]. We used a mutation rate of 1 substitution per 2565 years, this being the estimated time-dependent mutation rate close to the present, allowing for purifying selection [47]. The mutation rate following the correction curve over this interval changes by less than 7%, which should not affect the results in any significant way. This approximation is necessary as the founder analysis approach requires a linear mutation rate.

Following the analyses described above, we defined a new model with two periods of migration, based on the peaks of the first scans and archaeological/palaeontological evidence: one migration at 8 ka, corresponding to the peak of the Holocene Climate Optimum [20] to apportion prehistoric events, and a second migration near the present, at 0.5 ka, broadly corresponding to the beginning of the Atlantic slave trade but that statistically allocates all recent (historical) events. The resolution will not allow the separation of distinct very recent events (Atlantic slave trade and current migrants, for example), nor distinct potential early Holocene events, but it is nevertheless a valuable tool for demonstrating the statistical split between these two broad time periods. Following this split, we estimated the frequency of each proportion (lineages statistically allocated to recent events and to the early Holocene) in Europe and Southwest Asia. Frequencies were plotted geographically using the Kriging algorithm of the Surfer 8 software and considering only frequencies of sub-Saharan lineages (Appendix A) from full population data.

Finally, for each founder, we evaluated the phylogeographic patterns and defined a probable source location from among North, West, Central, Eastern, and Southern Africa, considering the geographic classification employed before [16].

## Figures and Tables

**Figure 1 ijms-23-09219-f001:**
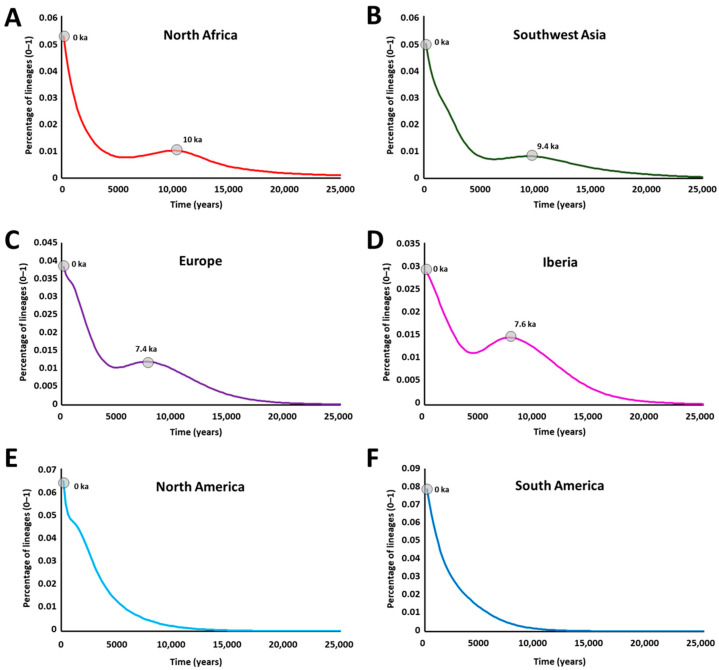
Probabilistic distribution of African mitogenome founder clusters across migration times over time, scanned at 200-year intervals from 0–25 ka, for North Africa (**A**), Southwest Asia (**B**), Europe (**C**), Iberia (**D**), North America (**E**) and South America (**F**). Point age estimates for each peak in ka are indicated.

**Figure 2 ijms-23-09219-f002:**
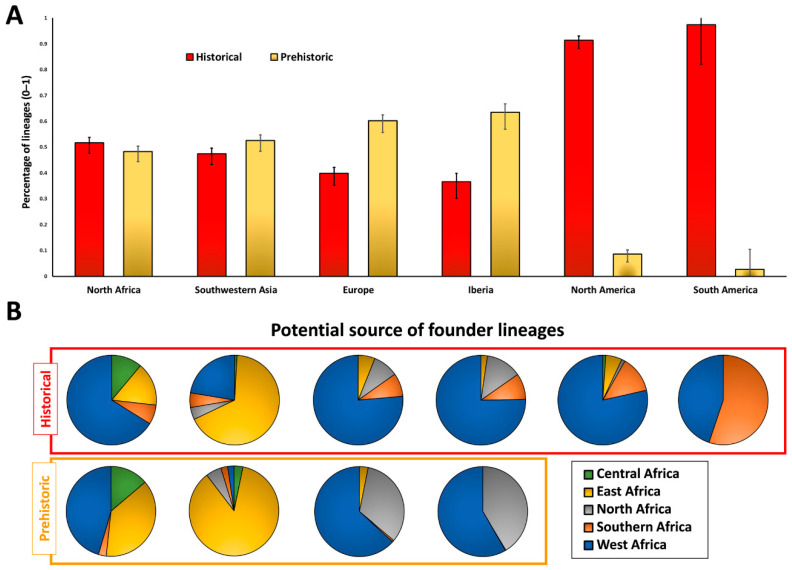
Founder analysis using a two-migration model for North Africa, Southwest Asia, Europe, Iberia, North America, and South America (**A**) and phylogeographic analysis of the most likely source of lineages for each migration (**B**).

**Figure 3 ijms-23-09219-f003:**
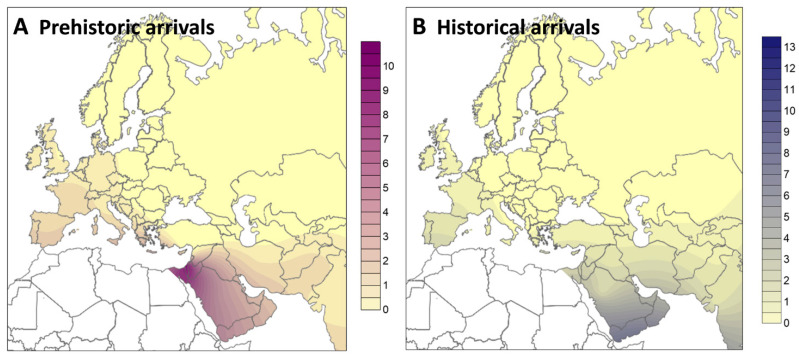
Frequency of sub-Saharan lineages in Europe and Southwest Asia following the statistical allocation into prehistoric arrivals (**A**) and historical arrivals (**B**). Plots estimated using the Kriging algorithm of the Surfer 8 software.

## Data Availability

Not applicable.

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
