# Peer review of "Phylogeography of Sub-Saharan Mitochondrial Lineages Outside Africa Highlights the Roles of the Holocene Climate Changes and the Atlantic Slave Trade"

_ijms, 2022, doi:10.3390/ijms23169219_

Round 1

Reviewer 1 Report

This is an interesting paper on the dispersal of African mitogenomes. The authors compiled mitogenome sequences from the literature and/or public databases, and using different models, looked into the founding time periods and haplotypes in various regions of the globe.

Can the authors provide some more details on the founder analysis method that they developed in their lab?

It would be very beneficial if the authors could include some maps to show the lineage distributions visually in this paper. These maps can be produced using a software program like Surfer. Including such figures within the main text (instead of supplemental material) would help the citation rate of this paper.

Author Response

We would like to thank the reviewer for the positive and useful comments.

Reviewer: Can the authors provide some more details on the founder analysis method that they developed in their lab?

Reply: Thank you for the comment. We added as much detail as possible on the methodology regarding the founder analysis, including:

“The software (which is under development) reads the entire phylogenetic tree in a .xml format. It searches for shared haplotypes between the hypothetical source and the hypothetical sink (either existing or inferred ancestral haplotypes). Following the detection of hypothetical founders, certain conditions need to be fulfilled to be considered a founder, as stipulated in its original conception [46].”

“Using this approach, we performed a scan on the statistical distribution of the lineages across a linear timeframe allowing the detection of the most probable periods of migration given the data. Such an approach should be further validated and improved using other source of information (namely from archaeology, history, paleoclimatology and linguistics), but the present study marks a step forward in this regard.”

“We calculated standard errors according to Saillard et al. [56], allowing the estimation of realistic errors in the Bayesian partitioning, as introduced before in the methodology [16].”

Reviewer: It would be very beneficial if the authors could include some maps to show the lineage distributions visually in this paper. These maps can be produced using a software program like Surfer. Including such figures within the main text (instead of supplemental material) would help the citation rate of this paper.

Thank you for your comment. We created, for Europe and Southwest Asia, a map displaying the frequency of the total lineages associated with the prehistoric movements and one for the lineages associated with the recent migrations. The results are displayed as figure 3 in the main text with some relevant appointments in the text. We also added to the methodology:

“Following this split, we estimated the frequency of each proportion (lineages statistically allocated to recent events and to the early Holocene) in Europe and Southwest Asia. Frequencies were plotted geographically using the Kriging algorithm of Surfer 8 software and considering only frequencies of sub-Saharan lineages (table S1) from full population data.”

Reviewer 2 Report

This is a very well done and extensive work which provides additional evidence of the migrations of African populations into Europe, West Asia and the Americas. They validated their founder analysis with the known slave trade into the Americas. There are several interesting observations in this work, all of them well validated by the data. One of the most interesting is the migration out of Africa in just two waves: one at about 8Ky and the other in recent times, with no relevant migrations in between. Also, the differential  distribution of populations in migrations in prehistorical and historical times out and within Africa are of relevance.

I consider this work of significant value to be publishes as is. The only point I would like to have a deeper discussion is the small bumps at about 1,000 years in the North American populations.

Author Response

We would like to thank the reviewer for the enthusiastic and useful comments.

Reviewer: I consider this work of significant value to be publishes as is. The only point I would like to have a deeper discussion is the small bumps at about 1,000 years in the North American populations.

Reply: We added to the discussion on the small bumps already present in the text (“However, some small bumps at about 1.0 ka most likely represent arrivals with the Atlantic Slave Trade, possibly conflated with a few earlier minor movements in the case of Europe, and probably artificially inflated due to the presence of a small number of sequencing artefacts, which would have a disproportionate impact at this timescale. It is also possible that the mutation rate correction for purifying selection [51] at this very recent time depth is insufficient.”) a new section discussing other possibility for the bumps and the need to deepen these studies with more sampling:

“Given the high number of founders detected in North America, it is even possible that this number is still underestimated, since while the main lineages were detected, some of the variation within those lineages might have already been carried to North America, slightly increasing estimated founder ages. Further sampling of African populations (especially West African populations) should help to settle this issue.”